# “Green” Biomaterials: The Promising Role of Honey

**DOI:** 10.3390/jfb12040072

**Published:** 2021-12-09

**Authors:** Gregorio Bonsignore, Mauro Patrone, Simona Martinotti, Elia Ranzato

**Affiliations:** DiSIT—Dipartimento di Scienze e Innovazione Tecnologica, University of Piemonte Orientale, Viale Teresa Michel 11, 15121 Alessandria, Italy; gregorio.bonsignore@uniupo.it (G.B.); mauro.patrone@uniupo.it (M.P.); simona.martinotti@uniupo.it (S.M.)

**Keywords:** green chemistry, honey, nanotechnology, silver nanoparticles, scaffolds

## Abstract

The development of nanotechnology has allowed us to better exploit the potential of many natural compounds. However, the classic nanotechnology approach often uses both dangerous and environmentally harmful chemical compounds and drastic conditions for synthesis. Nevertheless, “green chemistry” techniques are revolutionizing the possibility of making technology, also for tissue engineering, environmentally friendly and cost-effective. Among the many approaches proposed and among several natural compounds proposed, honey seems to be a very promising way to realize this new “green” approach.

## 1. Introduction

Natural products are useful to obtain and design drugs from ancient times. Resources from nature have been used as herbal medicine throughout human history [1]. Today, approximately half of marketed drugs are obtained from natural compounds. The use of herbal medicines, nutraceuticals, or phytonutrients is growing rapidly all over the world, also in several developed nations with complementary and alternative medicines (CAMs) approaches [2].

More in general, natural compounds or natural products are chemical complexes found in plants, animals, and micro-organisms, as well as synthetic or semi-synthetic compounds [3]. Several natural compounds show interesting biological or pharmacological activities, providing such beneficial effects for treating some human ailments. In the last decades, the interest in using natural products, from classical chemistry to drug discovery, is growing, in particular, to explore their potential against: microbial infections, cancer, inflammation, tissue damage, and other diseases [4].

New products are coming into the market, so public health questions and concerns surrounding the use of natural products are increasing their global use [5].

However, the success of natural products in clinical application and trials has been less impressive than expected, partly due to their low bioavailability, inappropriate molecular size, scarce water solubility, and low lipophilicity making structural instability in biological conditions, high metabolic rate, and rapid clearance [6].

Nanotechnology represents an interesting area to increase bioavailability, activities, and applications of natural products.

Moreover, the use of natural compounds is also increasing for wound healing and tissue regeneration practices. A damaged tissue/organ can be repaired using tissue-engineering approaches. Tissue-engineered 3D scaffolds can be employed as a temporary extracellular matrix to mimic the natural microenvironment, promoting cell adhesion and growth. These engineered scaffolds have been realized by biocompatible and biodegradable materials, including synthetic polymers and natural compounds.

## 2. Nanotechnology

The concept of nanotechnology started in 1959 and is related to a “There’s plenty of room at the bottom” lecture given by Richard Feynman at an American Physical Society meeting on 29 December in 1959. However, the word “nanotechnology” itself started to be used in 1974 when Norio Taniguchi used it to describe precision level in manufacturing materials at the nanometer scale.

In brief, nanotechnology is the ability to handle and control materials with atomic or molecular precision.

Nanotechnology has been used to improve the bioavailability, activity, and therapeutic index of various natural products. Furthermore, with recent nano-formulation, it is now possible to improve the properties and applications of natural compounds that in the past have proposed several shortcomings and limitations, such as very short circulating life. In chemotherapy, a natural compound can cause damage to both normal and cancerous tissues and cells. Designing a specific nano-carrier, a natural compound can be formulated for targeting mainly cancer cells [7].

Low bioavailability and poor solubility represent a relevant barricade for the employment of some natural substances. To this aim, specific formulation with nanomaterials can reduce premature drug loss, and bioavailability can be significantly improved.

A very illuminating example is the use of curcumin, a natural product of biological importance [8], badly absorbed by intestinal cells after oral administration due to curcumin’s low stability in gastrointestinal fluids as well as for its poor water solubility. These aspects restricted the possible application of curcumin. To avoid these limitations, curcumin has been encapsulated in liposomes, allowing better absorption by the intestinal tract [9]. In this manner, in a rat model of prostatic cancer, encapsulated curcumin inhibits carcinogenesis to a greater extent than free curcumin [10].

Another important plant-based compound with an increasing opportunity with nanomaterials is paclitaxel. Paclitaxel is a well-known anticancer compound, but its selective toxicity is increased when loaded in biodegradable nano-formulations based on linoleic and malic acid [11]. Moreover, Liang and coworkers [12] found that inhibition properties on HepG2 cells growth by the poly(gamma-glutamic acid)-poly(lactide)-paclitaxel nanoparticles was comparable to that of a clinically available paclitaxel formulation, while the nanoparticles showed significantly less activity.

Camptothecin, an alkaloid obtained from the Chinese tree *Camptotheca acuminate*, demonstrated strong anticancer activity, but it also has low solubility in water and in a physiologically well-tolerated solvent. Camptothecin, loaded in glycol chitosan nanoparticles, demonstrated higher stability, longer blood presence, and better accumulation in tumor tissues compared to free camphothecin [13].

## 3. Green Nanotechnology

Nanoparticles synthesis can be distinguished into two different kinds called top-down and bottom-up strategies.

Bottom-up or self-assembly technology uses chemical or physical forces (vapor deposition, laser pyrolysis, sol-gel processing, etc. [14]) operating at the nanoscale to assemble basic units into larger structures, while the top-down approach consists of breaking down of larger structures to obtain smaller ones.

Most of these approaches typically require the use of toxic and dangerous chemicals, for example, hydrazine, sodium borohydride, and dimethyl formamide, as well as drastic physical conditions such as vacuum and high temperature [15]. Moreover, these approaches may generate toxic products posing environmental biological risks, causing adverse effects on organisms (humans included) at various trophic levels [16,17,18].

Therefore, it seems mandatory to reduce the use of hazardous chemicals and to choose green methods for nanoparticles productions.

In fact, the latest developments in the nanotechnology fields focus on the application of cost-effective and environmentally friendly approaches. So, this “green nanotechnology” provides tools for the transformation of biological systems to green approaches to nanomaterial synthesis while preventing any associated toxicity.

Green science is helping to reduce the release of hazardous waste into the environment. The green approach also plays a crucial role in the production of fuel cells, the development of alternative energy science, and the building of batteries for storing energy, as well as new strategies to make solar cells [19,20,21].

Green nanotechnology includes various strategies such as the use of plant and microbial systems and, more in general biological approaches [22,23,24]. Some researchers have demonstrated the algae potential to produce gold, zinc, silver oxide as well as iron oxide nanoparticles [25,26]. In addition, bacteria and fungi have been used to synthetize some nanoparticles [27,28]. Moreover, root, seed, and leaf extracts were used successfully to produce nanoparticles [29]. Furthermore, other secondary metabolites with redox function, such as sugars (e.g., glucose and sucrose), flavonoids (e.g., luteolin and quercetin), and various amino acids (e.g., aspartate) have also been used successfully to realize metal nanoparticles [30,31].

Many bacteria and fungi have been used to synthetize suitable quality nanoparticles, but this approach requires a certain effort for cell culturing and harvesting. These methods often can involve contamination with microbial biomolecules. Plants have been considered a more environmentally friendly biological method for nanoparticles synthesis for their potential of biological reduction in metallic ions [32].

However, in the last decade, another renewable product, such as honey, emerged as an interesting option to produce metallic nanoparticles due to its compositions rich in various bioactive molecules, i.e., sugars, polyphenols, phenolic acids, proteins, able to reduce and stabilize the metallic ions [33,34].

Green technologies are also inducing a revolution in the biomaterials approach. In fact, new technologies and materials have been realized, reducing the synthesized scaffold use (such as cartilage, blood cell, bone, nerve, skin, etc.) and fabricating innocuous and eco-friendly assemblies for tissue engineering [35].

“Green techniques” have shown the feasibility to improve nanostructured scaffolds mimicking better the extracellular matrix (ECM) structure compared to the structures developed using macro/microfabrication techniques [36].

## 4. Honey

Since biblical times and before, honey showed a close history with humankind. Honey was not only the first sweetener used, but it was for a long time also intended as an ointment and a drug [37].

Honey, produced by bees from nectar collected from several plants [38], is a supersaturated mixture of sugars, with a presence in small quantities of other substances, flavonoids, phenolic compounds, organic acids, proteins, minerals, vitamins, very important for the properties of the honey itself [39,40]. Two monosaccharides, such as fructose and glucose, are the main components of honey, comprising about 95% of honey dry weight; the fructose-glucose ratio is above 0.9–1.5 [41]. Moreover, some authors have reported that glucose and fructose are also the main sources of reduction properties of honey. Both sugars act as reducing sugars; they react with Benedict’s (Cu^2+^ to Cu_2_O), Tollens’ (reduction in Ag^+^ to Ag^0^), and Fehling’s, or reagents [42]. All honeys show a pH value ranging between 3.5 and 5.5 due to the presence of organic acids (mainly gluconic acid).

Several positive effects can be attributed to honey. The best known is the antibacterial action, due to the production of hydrogen peroxide, to the large presence of sugars, and to some small compounds with antibacterial action [37,39].

In recent years, honey’s positive role in wound repair is also emerging. Martinotti and collaborators [43] showed the involvement of H_2_O_2_ as the main mediator of regenerative effects boosted by honey on skin cells. They demonstrated that H_2_O_2_ produced by honey in the extracellular space pass through aquaporin-3, the plasma membrane, reaching the cytoplasm and modulating the intracellular Ca^2+^ signaling and promoting wound repair (see Figure 1).

Honey also possesses some interesting anti-inflammatory characteristics making it very interesting for wound repair and tissue repair. The multifaceted possibilities offered by honey are currently used for scaffold applications in tissue healing [44].

## 5. Honey and Silver Nanoparticles

The positive effect of several metals and their salts has been described from the past [45]. The phoenicians used silver vessels to preserve wine, vinegar, and water, during their trips, while ancient Egyptians used silver powder to provide beneficial wound and tissue healing properties [46].

Since ancient times, silver has been attention-grabbing for its antibacterial properties for biomedical claims, water, and air purification, cosmetics, food production, and other household products. Silver has also been attractive for its low cost and abundant natural presence [47].

Topically applied silver sulfadiazine cream [48] is the standard antibacterial therapy for severe burn wounds and is still widely used [49]. However, low local retention and acute cytotoxicity effects have limited the clinical use of these ionic forms of silver materials [50]. Recent advancements in nanotechnology fields make it now possible to produce silver at the nanoscale level [51]. Silver nanoparticles (AgNPs) might also display supplementary antimicrobial capabilities not shown by ionic silver because of their large surface-to-volume ratio and small size, which induce physical and chemical changes in their assets [52]. AgNPs are still extensively examined due to their superior properties, such as chemical, physical, and biological characteristics, compared to their bulk forms [53].

AgNPs can be realized with different approaches, with many advantages and some limitations. The most common method of AgNPs production is through silver salt (e.g., AgNO_3_) chemical reduction using a reducing agent (e.g., sodium borohydride). During chemical reduction, silver ion (Ag^+^) receives an electron from the reducing agent and reverts to its metallic form (Ag^0^), clustering to form AgNPs [45].

A capping agent (e.g., poly(N-vinyl-2pyrrolidone)) is used during chemical reduction to stabilize the nanoparticles, also preventing them from aggregation [52]. However, reducing agents and other organic solvents used, such as sodium borohydride and N,N-dimethylformamide, are highly reactive, posing also important environmental and biological risks [52]. Moreover, the inability to control the AgNPs size produced by chemical reduction may result in a wide distribution of obtained AgNPs dimensions.

In the last years, many studies have been focused on developing new green chemistry methods for the synthesis of AgNPs with the advantage of using natural products and avoiding toxic organic solvents, reducing agents, and expensive purifications with high dangerous residuals [54].

Asmathunisha and Kathiresan [55] suggested that nanoparticle production could be induced by several compounds such as flavonones, amides, phenolics, amines terpenoids, alkaloids, proteins, and other reducing agents naturally existing in the plant extracts and microbial cells.

AgNO_3_ was added to chitosan extracted from some bacterial strains and then dissolved in acetic acid. The silver was then reduced through γ-radiation and stabilized by chitosan [56]. Some studies tried to produce AgNPs in the lamellar space of montmorillonite/chitosan by using the reduction approach by UV irradiation in the absence of any heat treatment or reducing agent [57].

Moreover, AgNPs have also been realized via AgNO_3_ photo-reduction in laponite, layered inorganic clay suspensions, which act as stabilizing agents preventing nanoparticles from aggregating instead of using chemical reagent [58].

The main disadvantage of using these green chemistry approaches is the risk of contamination due to the pathogenic bacteria generated during the purification [45].

So, essential oils and herbal extracts reached more attention as methods of green chemistry due to their antioxidant and antimicrobial properties, also avoiding pathogenic bacteria use for synthesis or extraction [59]. However, the main limitation of herbal extract use is again the wide distribution of particle sizes.

The special chemical characteristic of honey makes it very useful for nanoparticles with “green” synthesis. In fact, honey offers more advantages than bacteria-mediated methods; in particular, honey is a quick system compared to microbial approach and plant-based methods.

Despite honey having been proposed as a possible agent in the green synthesis of AgNPs, until now, only a few researchers have focused their attention on honey application for this purpose.

In 2009, a first paper analyzed the possible applications of honey for the synthesis of gold nanoparticles [31]. A total of 50 mg of HAuCl_4_ was dissolved in 120 mL deionized water, while 20 g of natural honey was diluted to 70 mL. The speed of reduction was found to increase with the increase in the addition of honey. Philip demonstrated [31] that this honey-mediated biosynthesis of Au nanoparticles was feasible, and the process showed economic viability. The nanoparticles obtained were characterized, showing a size ~15 nm, as demonstrated by transmission electron microscopy (TEM) image and X-ray diffraction (XRD) pattern. Moreover, this paper indicated as a possible reducing agent fructose and as a capping material for stabilization the honey proteins.

In 2010, Philip [60] reported the first simple and environmentally friendly method to produce very small silver nanoparticles in water at room conditions by means of natural Kerala honey as reducing and protecting agents. A total of 15 mL of this honey was added to 20 mL aqueous solution of AgNO_3_ (10^−3^ M) and stirred well for 1 min. The pH was adjusted to 6.5 using NaOH to start the reduction in Ag ions.

Crystalline silver nanoparticles (around 4 nm) generated at ambient conditions were stable for about 6 months. Glucose was proposed as a possible reducing agent, and the capping materials responsible for stabilization were proteins present in honey.

Sreelakshmi and coworkers [61] in 2011 developed a one-step green approach for gold and silver nanoparticles stabilization using natural honey at room temperature. This is the first report on silver and gold nanoparticles synthesis using natural honey, also exhibiting antimicrobial activity.

In particular, they suggested that poly-sugars naturally occurring in the honey perform both reduction and stabilization of metallic ions. Moreover, they also explored the antibacterial activity of these nanoparticles on some microbial reference strains, with encouraging results. The authors used a protocol slightly modified from Philip [31,60] 10 g of honey (information about the nature of honey was not available) was dissolved in 100 mL of distilled water. Then, the honey solution was added to aqueous HAuCl_4_ (10^−3^ M), allowing it to stand for 2 h at room temperature, during which reduction in Au^3+^ started. The efficiency of reduction was enhanced with increased honey concentration. TEM images determine the average size of nanoparticles around 9.9 nm. In the case of Ag nanoparticles, high dispersion levels were described as compared to that of Au nanoparticles. This may be attributed to the intrinsic properties of the two metals, such as surface energy, melting point, etc. The absence of Ag and Au agglomeration nanoparticles was due to the presence of functional groups in the honey, so anchoring the formed nanoparticles.

Obot and collaborators [62] in 2013 chose to use natural honey and sunlight irradiation for easy and fast synthesis of silver nanoparticles.

The honey attends as both reducing and capping agents without other stabilizing intermediates.

For the silver ions reduction, 5 mL of natural Nigerian honey was added to 95 mL of 0.1 M aqueous AgNO_3_. The stirred reaction mixture was exposed to bright sunlight, and after a few exposure seconds to sunlight, the solution turned yellowish-brown, indicating the formation of silver colloid.

These silver nanoparticles, stable for more than 6 months, showed excellent protection against steel against corrosion in acidic solution. As already observed, the authors proposed fructose as a possible reducing agent and honey’s proteins as capping material.

In 2013, Haiza and collaborators [63] prepared silver nanoparticles in a cost-effective, easy, and reproducible green system approach. Malaysian (Tualang) honey was used as stabilizing and reducing agent instead of hazardous compounds, such as dimethyl formamide and sodium borohydride. Honey dilution and pH solution were used to modulate the shape and size of nanoparticle formation. A total of 20 g of honey was dissolved in 80 mL of deionized water. Then, 15 mL of this honey was put in a 20 mL aqueous solution of AgNO_3_ (10^−3^ M). The pH was adjusted to 6.5 to initiate the Ag ions reduction. The reduction started promptly, as indicated by the golden yellow color of the solution. Based on scanning electron microscopy (SEM) results, the particles size decreased as the pH of the aqueous solution increased. So, the authors deduced that there was a rapid reduction in Ag ions and the formation of small nanoparticles at higher pH values. Moreover, the addition of NaOH, increasing the pH of the solution, produces an effect on the nanoparticle size, probably for the augmented formation of gluconic acid from glucose. Then, Ag ions oxidized glucose to gluconic acid and themselves reduced to metallic Ag. However, the authors did not recognize the ingredient(s) responsible for the reduction in the Ag ions. This work demonstrated that with honey was possible to produce Ag nanoparticles through green methodology, avoiding the presence of noxious solvent and waste.

AgNPs in stable colloidal solutions have been demonstrated to be effective against *Aspergillus* and *Penicillium* species [64]. In 2016, the inhibitory effect of AgNPs was also confirmed [65], derived by a safe, nontoxic method from Egyptian honey, on fungal growth as well as against ochratoxin A and aflatoxins produced by *Aspergillus ochraceus* and *Aspergillus parasiticus*. For the reduction in silver ions, 5 mL of 10% honey was added to 5 mL of aqueous AgNO_3_ solution. The reaction mixture was incubated at 30 °C for 72 h. The solution color changed to yellowish-brown, suggesting silver nanoparticles formation. The TEM analysis demonstrated that the size range of AgNPs was between 9 and 22 nm.

Another group [66] tested the efficiency of commercially available silver nanoparticles and the same silver nanoparticles supplemented with honey in *Aspergillus flavus* inhibition and aflatoxin B1 production prevention in the stored grains of maize. The outcomes indicated the effectiveness of the honey-enriched silver nanoparticles in inhibition of aflatoxin B1 compared to silver nanoparticles obtained by the classic approach, suggesting the importance of honey as a natural compound able to increase nanoparticles’ value.

González Fá and collaborators [67] studied the production of silver nanoparticles using low honey concentrations in acidic and alkaline media. A total of 25.0 g of honey (obtained from local markets from Buenos Aires, Argentina) was dissolved in 100.0 mL of water. An appropriate volume of this solution was added to 265 μL of AgNO_3_ solution (10^−3^ mol L^−1^), and the pH was adjusted at pH 5.0 and 10.0. Initially, this honey solution shows a very pale yellow color, while after stirring, it appears a typical intense brown-yellow color, indicating the formation of silver nanoparticles. The glucose in honey serves as a reducing agent, as well as a stabilizing agent. Fructose did not show an effect on silver nanoparticle formation. Interestingly, silver nanoparticles synthesized with glucose are larger and more diverse than those synthesized with honey, suggesting that the honey minor components have a significant role in nanoparticle formation. The synthesized nanoparticles, smaller than 20 nm, were obtained for the first time at pH 5.0. This synthesis was simple, at low cost, avoiding the use of hazardous and toxic substances, contributing to green chemistry goals. This process did not require washing, filtering, heating, sonication, or microwave treatment.

Al-Brahim and Mohammed [68] tested the use of honeys from two different floral origins (*Acacia gerrardii* and *Ziziphus spina-christi*) as agents to carry out the production of AgNPs, also evaluating their antioxidant, antibacterial and cytotoxic effects. For the biogenic synthesis of AgNPs by Ag ion reduction, 1 mL of honey was added to 5 mL AgNO_3_ (1 mM). The solution was homogenized and incubated under dark conditions for 72 h at 30 °C. Conversion of the mixture color to brown was considered to be the first indicator for AgNPs formation. These honey-prepared AgNPs possess a cytotoxic effect against the HepG2 cell line and positive action on some Gram-positive and -negative bacteria strains. However, the exact mechanisms of action should be further characterized.

Taken together, the observations and data about honey use for AgNPs synthesis showed that nanoparticles synthesized using honeys that themselves possess antimicrobial properties displayed better synergistic or additive effects than nanoparticles alone.

The main limitation of these studies, which make it very difficult to compare results, is related to different honey types used as well as the use of honey not standardized or not also characterized in term of antibacterial properties.

So, further researches focused on the honey concentration and honey type to use for the “green-synthesized” AgNPs are strongly advisable. Additional studies are also necessary to establish for AgNPs safety and other biological effects than antimicrobial activity.

## 6. Honey and Other Nanoparticles

Recently, Neupane and collaborators [69] studied the loading of Himalayan honey into iron oxide nanoparticles (IO-NPs) and their antibacterial and antioxidant properties.

IO-NPs consist of maghemite (γ-Fe_2_O_3_) and/or magnetite (Fe_3_O_4_) particles with diameters ranging from 1 to 100 nm. Unlike other metallic analogs, IO-NPs can help to fight diseases and infections for their well-known biocompatible and magnetic properties.

The honey used for this application was collected from the high-altitude regions of the Kaski district located in western Nepal, where diverse medicinal plants grow.

IO-NPs were obtained from FeCl_3_ 2H_2_O and FeSO_4_ 7H_2_O by grinding in the presence of sodium citrate as a complexing agent and stabilizer to avoid particle aggregation. The average size of honey-loaded IO-NPs was found around 33–40 nm, with a more needle shape compared to free IO-NPs, suggesting the surfactant-like behavior of honey. Ultraviolet-visible (UV-VIS) spectroscopy of both free and honey-IO-NPs confirmed the honey loading onto nanoparticles. The antioxidant activity of Himalayan honey can be due to antioxidants’ presence, such as phenolic and flavonoid compounds, as already demonstrated for other honeys in several models [39,44]. Moreover, Fe_3_O_4_ nanoparticles also possess significant scavenging ability against the free radicals. This property could be attributed to the electron transfer from the Fe^+2^/Fe^+3^ systems of IO-NPs. As a result, the antioxidant activity of IO-NPs loaded with honey was increased two-fold in comparison to Himalayan honey alone.

Moreover, the results showed that IO-NPs loaded with honey displayed a synergistic effect on microbial growth inhibition and free radical scavenging activity, with interesting appeal for applications in the biomedical field. In conclusion, IO-NPs loaded with honey could represent a promising alternative as antimicrobial and antioxidant agents.

Other kinds of nanoparticles, i.e., copper nanoparticles (Cu-NPs), also possess unique characteristics making them very interesting, such as low cost, less toxicity, heat transfer properties, and high surface area to volume ratio. These characteristics are mainly due to their physical properties such as size, shape, morphology, composition, and crystalline phase. Cu-NPs can be used in several applications, such as for catalytic and energy applications [70] as well as for drug delivery and for their antibacterial properties. There are some classic strategies to synthesize Cu-NPs using physical and chemical methods. However, the major problems are the production of some toxic by-products, high cost, and laborious processing. Moreover, copper is highly unstable, and it can oxidize at atmospheric conditions. The honey composition can help to stabilize these nanoparticles. Nur Afini Ismail and colleagues [70] studied honey-mediated Cu-NPs synthesis assisted by ultrasonic irradiation. Honey (obtained from flowers of Australian eucalyptus and ground flora) was used as a reducing and stabilizing agent with the ascorbic acid (vitamin C) as a supporting reducing agent.

The Cu-NPs were realized using ultrasonic irradiation reduction. A total of 10% *w*/*v* honey was added into the copper nitrate solution and stirred until homogenization. After that, 0.6 M sodium hydroxide was added, reaching pH 7.5. Then, 15 mL of ascorbic acid (1 M) was added into the mixture solution and underwent ultrasound irradiation for 10 min. At the end, the solution was washed with water and dried in an oven. The obtained nanoparticles were more effective toward Gram-positive compared to Gram-negative bacteria. The potential biomedical application of Cu-NPs was also assessed by cytotoxicity assay against two mammalian cell lines.

Carbon nanoparticles can be incredibly useful in several fields, and they trigger the real field of nanotechnology due to remarkable physical and chemical properties, such as biocompatibility, inertness, and photoluminescence.

Wu and collaborators [71] derived naked carbon nanoparticles from commercial food-grade honey. Commercial food-grade honey (Great Value™ Clover Honey, Wal-Mart Stores, Inc, USA; batch composition—fructose: water: 17.2%, glucose: 31%, fructose: 38%, maltose: 7.1%, sucrose: 1.3% higher sugars: 1.5%) was suspended with an organic macromolecular passivating agent, purged with argon and heated for 30 min in a domestic microwave oven. The product changed from light yellow to dark brown to black. The obtained nanoparticles were purified by repeated centrifugation in water. Surface-coated particles are significantly smaller than the previously explored particles (gold, copper, etc.) for sentinel lymph nodes (SLN) imaging. These results indicate an interestingly rapid signal enhancement of the SLN with honey-mediated nanoparticles.

Among the different nanoparticle types, magnetic nanoparticles have been extensively explored for important biomedical and environmental applications such as drug delivery, cell labeling, cancer hyperthermia therapy, magnetic separation, and enhanced magnetic resonance imaging.

Several methods have been yet proposed for the synthesis of magnetite nanoparticles (Fe_3_O_4_ NPs), including chemical, physical and biological approaches. Rasouli and coworkers [42] used natural honey (from cold highlands of Northeastern China) to produce at room temperature, in a fast precipitation method, Fe_3_O_4_ NPs using iron (III) chloride, iron (II) chloride, and sodium hydroxide. This work suggested the applicability of green synthesis of super-paramagnetic Fe_3_O_4_-NPs using natural honey. The authors also proposed the fructose present in the honey as a possible co-reducing agent and natural honey proteins as the capping agent for stabilization. The TEM data demonstrated that the particle size was reduced from 3.21 to 2.22 nm with the honey increase respectively from 0.5% to 3.0% (*w*/*v*). In vitro cell viability of Fe_3_O_4_-NPs assessed by MTT assay on mouse fibrosarcoma cells showed any cytotoxicity up to 140.0 ppm, suggesting their safe use in biological applications such as drug delivery, for more details Table 1.

## 7. Honey-Based Scaffolds

Scaffolds are a great alternative to help wound healing. In fact, they can be used to obtain an optimal repair (reconstruction of derma, re-epithelialization, formation of granulation tissue) instead of fibrotic healing.

Scaffolds can have different characteristics, and researchers must choose the best by selecting their geometries and physic-chemical properties. The application of them varies with the changing of fabrication parameters.

Often beehive product-infused or impregnated biomaterial are made with gels or fibers obtained through electrospinning. While bone, cartilage tissue, and muscle are treated with hydrogel or cryogel-scaffolds for skin-related healing electrospun scaffolds are more often applied [72].

Scaffolds can be obtained from different natural materials:Silk fibroin (SF), a structural protein derived from the cocoon of *Bombyx mori*, is used to produce porous scaffold and nanofibers [73] useful for treating bone and cartilage [74,75];Chitosan (CH), derived from deacetylated chitin, is used for its important antibacterial properties and its ability to promote the release of extracellular matrix.

Pure honey cannot be added to cells and tissues; in fact, it may be toxic [44]. However, its use within the scaffolds can guarantee a safe and controlled release of its bioactive compounds.

### 7.1. Gel Scaffolds

Hydrogels are composed of networks of crosslinked polymers that can store water in large quantities [76]. They can be distinguished in physical gels and covalently crosslinked gels [77]. While cryogels, despite the similarity to hydrogel, are made through a quickly freezing and thawing, which generate a macroporous spongy-like structure [78].

A work realized in 2016 [79], starting with the research of new antimicrobial strategies for wound recovery, focused on the improvement of antibacterial activity combining different kinds of honey and gelatin included in hydrogel films composed of chitosan and PVA (poly-vinyl alcohol).

Thyme honey has shown a substantial upgrade on the antibacterial functions of the hydrogel films against *Pseudomonas aeruginosa* and *S. aureus*. While chitosan, PVA, and honey change the antibacterial activity (for both bacteria) according to their concentration, gelatin concentration is not related to the antibacterial effect. Changes in chitosan or honey seem linked to the swelling property; in fact, experimental tests based on different honey concentration shows a growing of mechanical strength and a reduction in water absorption, while chitosan regulates the swelling extent.

Scaffold based on manuka honey (MH) and gellan gum was studied to promote cartilage healing. The scaffold is absolutely not cytotoxic and allows hMSCs (human mesenchymal stem cells) to adhere and, after 21 days of incubation, cells differentiate in chondroblast. Moreover, they show antibacterial activity versus *Staphylococcus aureus* and *Staphylococcus epidermidis* [80]. The same authors described different possible scaffold combinations of MH (mesoporous silica, resveratrol, or silk fibroin).

A hydrogel sheet composed of chitosan, sunflower honey (20%), and bovine gelatin (supplementary) shows a rabbit skin burn healing time faster than the commercially available ointment or the controls. Moreover, it is effective against *Escherichia coli* and *Staphylococcus aureus* [81].

In recent work, a combination of chicory honey (0–20% *v*/*v*) with a polymer solution (chitosan, PVA, and gelatin = 2:1:1) shows antibacterial function (*Staphylococcus aureus* and *Pseudomonas aeruginosa*) and promotes both fibroblasts proliferation in vitro and wound closure in a rat model [82].

In another work, scientists focused on a combination of 2–10% honey and a hydrogel sodium alginate-based. The increase in honey concentration is related to the decrease in stiffness (according to the investigators, amylase and glucosidase contained in honey might be responsible for undermining covalent bonds resulting in the decrease in this parameter) and hydrophilicity; moreover, it is correlated with the increase in swelling index. The in vivo analysis showed the best antimicrobial activity and wound healing at a 4% concentration of honey [83].

Some topical hydrogels were obtained from the combination of honey and CH or carbopol 934. They are very effective on burn infection bacterial strains [84].

Carboxymethyl cellulose and chestnut honey were evaluated as competitive options as adjuvant therapies of diabetic ulcer wounds [85]. The percentage of honey in these hydrogels is correlated to water uptake and antimicrobial function.

An interesting scaffold is obtained from the combination between pectin and MH. Pectin is a heterosaccharide contained in plant cell walls, which has a protective role. It is studied for drug delivery and scaffolding. Pectin-MH hydrogel is not cytotoxic, can absorb water easily, and has shown antibacterial properties [86,87].

A particular hydrogel is composed of PEGDA (poly-ethylene glycol-diacrylate) and a honey-inspired peroxide-producing enzyme called glucose oxidase. This hydrogel was made for industrial application; however, it has shown antimicrobial function against *S. epidermis* [88].

### 7.2. Electrospun Scaffolds

Currently, a new method to make honey scaffolds (for wound healing application) is the incorporation into electrospun fibers. Its success in application is due to different elements such as structure, flat geometry, mechanical properties, and watery consistency.

The electrospinning technique involves the pushing of a polymer solution with a syringe and then the application of a high-intensity electric field, a conductive needle [78].

A voltage is applied at the needle tip, the solvent evaporates, and the polymeric scaffold is collected in the grounded target [89]. However, sometimes it is required a toxic solvent.

A biodegradable and synthetic polymer called poly (1,4-cyclohexane dimethylene isosorbide trephthalate) or (PICT) is commonly used. PICT and honey were used in three blend ratios (90:10, 85:15, and 80:20) for developing electrospun nanofibers with antimicrobial characteristics.

Among them, the second ratio is the most suitable option; in fact, it shows the best elastic behavior and tensile strength (increasing honey quantity is related to larger diameters and hydrophobicity decreasing).

MH is a cross-linking agent; in fact, it can be used to fabricate fiber mats of PCL or a mix of PCL and methylcellulose (MC) [89]. Thanks to their biodegradability, these fibers are considered useful to provide bioactive glass particles (BG). PCL fibers, including manuka honey and bioactive glass particles, improve the release rate of honey. On the contrary, the blend fibers show less effect. MH causes an increase in fiber diameter and a reduction in tensile strength of fibers. However, it does not change fibroblasts’ viability in mixed fibers but improves the migration of keratinocytes. Antibacterial effects have not been found [78].

In recent times, Hixon et al. focused on the connection between the antibacterial effect of MH and the shape of silk fibroin-based scaffolds [90]. MH is released by electrospun scaffolds faster than hydrogels and cryogels. As a result, *S. aureus* improves its clearance and adhesion. The antibacterial effect of manuka honey depends on the concentration and the kind of bacterial colony. Various human endothelial primary cells and fibroblasts are cultured on PCL biomimetic nanofibers coated with manuka honey [91].

Honey concentration influences their variability; in fact, some honey concentrations can increase cell proliferation than the clean PCL mesh.

SF nanofibers are constituted by poloxamer 407 or manuka honey. If compared to each other, human dermal fibroblasts have a higher density in the honey scaffold [92].

Recently, the incorporation of manuka honey (1–20% *v*/*v*) in PCL nanofiber scaffold has been studied, and scientists showed how it could enhance the fibroblast infiltration into the scaffold and proliferation [93]. The scaffold degradation rate does not increase with honey incorporation; moreover, some of them contrast *Escherichia coli* growth.

### 7.3. Other Scaffolds

Another kind of scaffold used in tissue engineering is foams. Foams are freeze-dried, and they are a crosslinked combination between methylcellulose and manuka honey [94].

Honey reduces the contact angle, which causes an improvement in hydrophilicity and wettability. Normal and cell line-derived fibroblasts cultured with these scaffolds have viability like controls. The results of a scratch assay showed both the antimicrobial activity versus *Escherichia coli* and *Staphylococcus aureus* and the migration ratio of keratinocytes-like cells.

Honey used at 1% to 6% (*v*/*v*) concentration allows obtaining stable foam scaffolds. The rise of honey concentration causes the increase in pore size, swelling ratio, degradation; on the contrary, the pore-density decreases. In addition, honey promotes primary fibroblasts proliferation. In vitro studies have shown how a 4% honey concentration is the best to enhance the synthesis of ECM proteins by the cells infiltrated with scaffolds. In vivo, the 4% honey scaffold showed similar results; in fact, in the animal model, it speeded up the healing of the wound.

Currently, this kind of scaffold is the best contender for skin repair; in fact, it shows minimal scarring, physiological epithelialization, hair follicles, and blood vessels defined.

Another example is a manuka honey-PVA-based dressing, which can enable a prolonged release of honey and works as an antimicrobial and fibroblast proliferator. In addition, erythromycin can be loaded into the dressing, and honey does not affect the release of antibiotics [95], maintaining its effectiveness against *Staphylococcus aureus*.

A hopeful method to fabricate a blended scaffold is bioprinting. The 3T3 fibroblast proliferation is improved by a load of honey into printed scaffolds in the presence of a blended alginate scaffold in comparison to pure alginate [88].

These results of scaffolds containing honey are very promising, but further studies are strongly needed to be carried out regarding the honey type to use as well as to control the release behavior of honey. Alternative strategies to encapsulate honey are required to obtain a sustained release and therefore to open to realize long-term antimicrobial effects.

## 8. Conclusions

The development of ecological and cost-efficient synthesis approaches of nanomaterial production still remains a scientific challenge, very attractive because metal nanoparticles are used for various catalytic applications as well as for biomedical purposes.

Biological approaches for producing nanoparticles are increasingly considered, representing a green approach aimed to minimize environmental impact. In fact, the synthesis of nanoparticles requires three main steps: a metal salt, a reducing agent, and a stabilizer or capping agent.

Biosynthesis routes provide nanostructures with defined morphologies and sizes. Among the organisms reported for green chemistry, the preparation of bacteria is generally more expensive than the plant extracts preparation. Plants are interesting as an economic and valuable alternative for the large-scale biosynthesis of nanoparticles.

However, in the last decade, it is emerging the green synthesis of nanoparticles mediated by honey. Some properties make honey very attractive. Biomolecules present in honey can be used to reduce ions to nanoparticles, preparing particles free of contaminants, such as those used for therapeutic applications. Moreover, honey is readily available, nanoparticles produced are stable, and their rate of synthesis is comparable to classic methods, representing a real alternative for large-scale production. Honey sugars and other components may act as both reducing and capping agents in nanoparticles synthesis.

Taken together with data available, the presence of a broad variability of biomolecules in honey increases the rate of metal ions reduction, formation of nanoparticles, and their stabilization. Honey-mediated nanoparticles synthesis does not require high temperature and does not produce any harsh compounds, making the process suitable for not-sophisticated laboratories, also avoiding facilities required for cell cultures, which are essential when working on unicellular organisms.

Moreover, another great honey advantage is to be an absolutely sustainable energy source, available all over the world [37]. Green methods should be extended to produce other complex nanoparticles such as TiO_2_, Fe_3_O_2,_ and ZnO.

The main limits of honey use for nanoparticles synthesis are the strong necessity to identify the molecules present in honeys responsible for metal reduction, as well as to identify among the huge number of available honeys, the best(s) to generate nanoparticles.

Moreover, there is also a strong need to identify the most abundant honey proteins responsible for the stabilization of metal ions in nanoparticle production, as well as to understand the choice of correct honey for nanomaterial and scaffold use.

The main advantage of honey use is the low cost, in particular in developing countries, compared to similar synthetic products [96].

However, honey variation is great and depends on many factors, such as the seasonal collection time and botanical source.

Furthermore, the engineered scaffolds containing honeys have confirmed the maintenance of honey efficacy against bacteria. These results are based on in vitro data, and more observations are strictly required to confirm the behavior in an in vivo environment. Alternative strategies for honey encapsulation should be explored to achieve a sustained release opening the chance of realizing long-term antibiotic results.

Currently, scaffolds containing honey as well as honey-loaded nanoparticles are very popular, but relatively few studies have until now investigated the potential of honey because of its variability, as mentioned above. Standardization of the levels, at least of the antimicrobial compounds, is requested to the applicability of honeybee products for “green chemistry” purposes.

Moreover, the use of standardized preparation or isolated honey components would positively contribute to the level of control of biophysical and biological modifications of engineered materials. We envisage honey-mediated supports could find in the next future new applications in several fields such as coatings, imaging, and cellular and molecular biology.

## Figures and Tables

**Figure 1 jfb-12-00072-f001:**
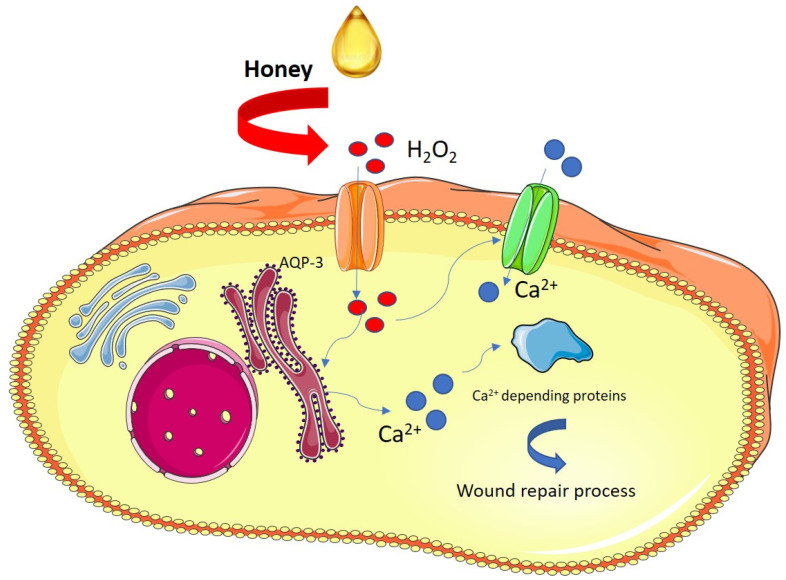
Schematic representation of the honey mechanism of action in the wound healing process. The schematic art pieces used in this figure were provided by Servier Medical art (https://smart.servier.com/ accessed on 17 November 2021).

**Table 1 jfb-12-00072-t001:** Researches realized on honey-based nanoparticles.

Application	Honey Type	Reducing Capping Agent	Reference
Honey-facilitated synthesis of gold nanoparticles	Kerala natural honey	Glucose Honey proteins	[31]
Honey-facilitated synthesis of silver nanoparticles	Kerala natural honey		[60]
Gold and silver nanoparticles stabilization using natural honey	Not specified (from India)		[61]
Sunlight-mediated synthesis of silver nanoparticles using honey	Nigerian honey	Fructose	[62]
Synthesis of silver nanoparticles using local honey	Malaysian (Tualang) honey	Honey	[63]
Honey-mediated silver nanoparticles	Egyptian honey		[65]
Honey-enriched silver nanoparticles in the detoxification of aflatoxin	Not specified		[66]
Synthesis of nanoparticles using low honey concentrations	Honey from local markets from Buenos Aires	Glucose	[67]
Synthesis, characterization and antioxidant and antimicrobial activities of Himalayan honey-loaded iron oxide nanoparticles	Himalayan honey		[69]
Properties of silver nanoparticles synthetized with Saudi Arabia honey	Saudi Arabia honey		[68]
Copper nanoparticles synthetized using honey	Honey from flowers of Australian eucalyptus and ground flora		[70]
Derived carbon nanoparticles from honey	Clover honey		[71]
Honey-based synthesis of Fe_3_O_4_ nanoparticles	Honey from cold highlands of Northeastern China	Fructose	[42]

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
