# Peer review of "“Green” Biomaterials: The Promising Role of Honey"

_jfb, 2021, doi:10.3390/jfb12040072_

Round 1

Reviewer 1 Report

Congratulations to the authors for the great effort to review such a topic. The only problem I perceive is there is already a review from 2017, which includes many of the subjects discussed within this manuscript, but scaffolds. Maybe some other applications different from scaffolds but also interesting in the field could be included in order to provide a stronger contribution to the field.

Concerning some aspects close to nanotechnology, I think there are two questions that should be clarified:

  1. Line 45: The definition of nanotechnology here provided might give the impression that only controlling molecules/atoms is nanotechnology, but it is a wider term. I suggest rewriting the sentence this way: “ability to handle and control materials with atomic or molecular precision”. This way it seems clearer that no matter the size of the object to handle, it will be nanotechnology if you do it atomically, or molecularly.
  2. Line 76: There is no description for bottom-up approach, when top-down has just been described. Therefore, I suggest explaining it, instead of citing examples of bottom-up technologies.

English language is perfect, though there are some minor spelling mistakes, and some sentences that could easily be improved. Please, let me point out some of them:

  1. Line 13: Unfortunately or however?
  2. Line 16: Cost-effective
  3. Line 29-30: Simplify/clarify rewriting “the interest for…their potential against…”
  4. Line 32-33: Clarify rewriting “as the global use of them… market”
  5. Line 177: Clarify “avoiding also… utilization”. I think you have mixed the concept of avoiding the risk of contamination due to pathogenic bacteria, and at the same time the need of using bacteria. Rewriting the sentence should address this issue.
  6. Line 317: There are two “imaging” words, and I guess the first one is not correct.
  7. Line 353: I guess the authors wanted to say “But its use within scaffolds…”
  8. Line 358: Distinguished
  9. Line 487: “A strong need TO identify…”

Author Response

Congratulations to the authors for the great effort to review such a topic. The only problem I perceive is there is already a review from 2017, which includes many of the subjects discussed within this manuscript, but scaffolds. Maybe some other applications different from scaffolds but also interesting in the field could be included in order to provide a stronger contribution to the field.

We thank reviewer for taking time to comment our ms and for the positive evaluation.

We have modified the text according to suggestions.

Concerning some aspects close to nanotechnology, I think there are two questions that should be clarified:

  1. Line 45: The definition of nanotechnology here provided might give the impression that only controlling molecules/atoms is nanotechnology, but it is a wider term. I suggest rewriting the sentence this way: “ability to handle and control”. This way it seems clearer that no matter the size of the object to handle, it will be nanotechnology if you do it atomically, or molecularly.

We have modified the sentence according to suggestion.

  1. Line 76: There is no description for bottom-up approach, when top-down has just been described. Therefore, I suggest explaining it, instead of citing examples of bottom-up technologies.

We have modified the sentence according to suggestion.

English language is perfect, though there are some minor spelling mistakes, and some sentences that could easily be improved. Please, let me point out some of them:

  1. Line 13: Unfortunately or however?
  2. Line 16: Cost-effective
  3. Line 29-30: Simplify/clarify rewriting “the interest for…their potential against…”
  4. Line 32-33: Clarify rewriting “as the global use of them… market”
  5. Line 177: Clarify “avoiding also… utilization”. I think you have mixed the concept of avoiding the risk of contamination due to pathogenic bacteria, and at the same time the need of using bacteria. Rewriting the sentence should address this issue.
  6. Line 317: There are two “imaging” words, and I guess the first one is not correct.
  7. Line 353: I guess the authors wanted to say “But its use within scaffolds…”
  8. Line 358: Distinguished
  9. Line 487: “A strong need TO identify…”

We have corrected the mistakes or better clarified the sentences.

Reviewer 2 Report

'Honey and nanoparticles: an interesting link for a 2 green approach' is an interesting read on the application of honey-based composites. To help increase its impact and understanding to the readers, I have a few suggestions:

  1. The manuscript completely lacks figures from the cited research and schematics. A review can turn quite boring without any visuals. Also, it helps the reader to envision what the cited researchers have accomplished.
  2. The title does not do justice to the subjects covered in the manuscript since the review encompasses various physical forms (foams, fibers, hydrogels), besides nanoparticles.
  3. The authors should provide a critical overview of the merits and demerits of each approach at least at the end of the corresponding section.
  4. The authors need to remove grammatical mistakes and also merge some single-sentenced paragraphs.

Author Response

'Honey and nanoparticles: an interesting link for a 2 green approach' is an interesting read on the application of honey-based composites.,

We thank the reviewer for the positive evaluation of our ms.

To help increase its impact and understanding to the readers I have a few suggestions:

  1. The manuscript completely lacks figures from the cited research and schematics. A review can turn quite boring without any visuals. Also, it helps the reader to envision what the cited researchers have accomplished.

We have inserted Figure 1 to explain mechanism of action of honey in inducing wound healing process.

  1. The title does not do justice to the subjects covered in the manuscript since the review encompasses various physical forms (foams, fibers, hydrogels), besides nanoparticles

We have modified the Title.

  1. The authors should provide a critical overview of the merits and demerits of each approach at least at the end of the corresponding section.

We have modified the text with a more critical perspective.

  1. The authors need to remove grammatical mistakes and also merge some single-sentenced paragraphs.

We have corrected the mistakes and improved the English quality

Reviewer 3 Report

The review presented by Ranzato and collaborators explores the use of honey as a green approach for the synthesis of nanoparticles, with special emphasis on silver nanoparticles, as well as, for the improvement of scaffolds. Although it mentions most of the work done on these topics, the review lacks a critical discussion and seems more like a list of studies.

My detailed comments are provided below.

The title of the review may create confusion as it does not only describe the synthesis of nanoparticles, but also the use of honey for scaffolds.

The introduction is focused on the use of nanotechnology to improve the properties of natural products (mainly for medical or clinical applications), while the review is focused on the use of natural products (i. e. honey) to improve the properties of nanoparticles and scaffolds. Therefore, the introduction does not sound appropriate, and a more engaged introduction could help the readers to focus on the actual topic discussed on the review. For example, a more complete about green nanotechnology (point 3) could be more appropriate.

There are problems with references that are difficult to follow. For example, ref 16 (line 81) is about NPs and not about the toxicity generated by the nanoparticle synthesis methodologies stated in the text. References 17, 18, 27 and other are also wrong as they describe topics different to the ones describes in the text. Therefore, a careful revision of all the references should be carried out.

Description of the experimental details regarding the synthesis of nanoparticles (especially silver) with honey in sections 5 and 6 do not contribute to the review unless a compressive comparation is carried out as most of them seems very similar. In addition, if included, experimental details should be included for all the nanoparticles discussed, and not for some of them. For example, the synthesis of IO-NPs is not detailed but it does for Cu-NPs. Discussions about the role of honey or its proteins and sugars as stabilizing and reducing agents could be more beneficial.

In line with last discussion, Table 1 is very limited. To make a more critical discussion, a comparative table with the kind of nanoparticle, the size, the kind of honey, the reducing-capping agent (if discussed), use of additives, kind of honey, real applications (i.e. antimicrobial application of NPs…), etc, should be included as it would help the reader to detect strengths and weaknesses of the different approaches discussed in the text.

Finally, scaffolds can be or not related with nanotechnology and, although the literature review appears very interesting, it seems that it not fit with the aim of this review. If authors want to include this section in the review, a change in the title, abstract and introduction that explains other uses of honey, rather than synthesis of nanoparticles, is recommended. Again, a more critical discussion of the literature with short conclusions or appreciations would be value so the article does not read like a collection of studies.

Conclusions are also very relaxed and not totally coherent and supported by the listed citations.

In addition, English should be carefully revised.

In general, the review needs major revisions to be considered for its publication.

Author Response

The review presented by Ranzato and collaborators explores the use of honey as a green approach for the synthesis of nanoparticles, with special emphasis on silver nanoparticles, as well as, for the improvement of scaffolds. Although it mentions most of the work done on these topics, the review lacks a critical discussion and seems more like a list of studies.

We thank reviewer for taking time to comment our ms and for the suggestions to improve the ms.

My detailed comments are provided below.

The title of the review may create confusion as it does not only describe the synthesis of nanoparticles, but also the use of honey for scaffolds.

We have modified the Title of our ms.

The introduction is focused on the use of nanotechnology to improve the properties of natural products (mainly for medical or clinical applications), while the review is focused on the use of natural products (i. e. honey) to improve the properties of nanoparticles and scaffolds. Therefore, the introduction does not sound appropriate, and a more engaged introduction could help the readers to focus on the actual topic discussed on the review. For example, a more complete about green nanotechnology (point 3) could be more appropriate.

We have modified the Introduction Section considering the application for tissue engineering.

There are problems with references that are difficult to follow. For example, ref 16 (line 81) is about NPs and not about the toxicity generated by the nanoparticle synthesis methodologies stated in the text. References 17, 18, 27 and other are also wrong as they describe topics different to the ones describes in the text. Therefore, a careful revision of all the references should be carried out.

We have carefully revised the citations and inserted new ones.

Description of the experimental details regarding the synthesis of nanoparticles (especially silver) with honey in sections 5 and 6 do not contribute to the review unless a compressive comparation is carried out as most of them seems very similar. In addition, if included, experimental details should be included for all the nanoparticles discussed, and not for some of them. For example, the synthesis of IO-NPs is not detailed but it does for Cu-NPs. Discussions about the role of honey or its proteins and sugars as stabilizing and reducing agents could be more beneficial.

We have better discussed the silver nanoparticles pro and cons in the section 5 as well as we have better detailed the IO-NPs synthesis (section 6).

In line with last discussion, Table 1 is very limited. To make a more critical discussion, a comparative table with the kind of nanoparticle, the size, the kind of honey, the reducing-capping agent (if discussed), use of additives, kind of honey, real applications (i.e. antimicrobial application of NPs…), etc, should be included as it would help the reader to detect strengths and weaknesses of the different approaches discussed in the text.

We have expanded Table 1 information.

Finally, scaffolds can be or not related with nanotechnology and, although the literature review appears very interesting, it seems that it not fit with the aim of this review. If authors want to include this section in the review, a change in the title, abstract and introduction that explains other uses of honey, rather than synthesis of nanoparticles, is recommended. Again, a more critical discussion of the literature with short conclusions or appreciations would be value so the article does not read like a collection of studies.

We modified title, abstract and introduction section according to reviewer’ suggestions.

Conclusions are also very relaxed and not totally coherent and supported by the listed citations.

We have modified the Conclusions.

In addition, English should be carefully revised.

We have corrected the grammar errors and improved the English quality.

In general, the review needs major revisions to be considered for its publication.

Round 2

Reviewer 3 Report

Dear Authors,

Thank you for your reviewed manuscript. 

I think your work on it have improved its quality, so I agree to publish it. 

My only concern regards to table 1. You have considered AgNO3 or HAuClas additives in the reaction synthesis, while they are the main reactive for the synthesis of nanoparticles. Therefore, they can not be considered as additive. I would remove this column about additives in table 1 if you feel there is not enough information. 

Kind regards,

Author Response

We thank reviewer for taking time to comment our ms and for the positive evaluation.

We have modified the table according to suggestions, so we removed the column about additives.